# OpenReview forum: "POROver: Improving Safety and Reducing Overrefusal in Large Language Models with Overgeneration and Preference Optimization"
_ICLR.cc/2025/Conference — Submitted to ICLR 2025_

### Official Review · Reviewer_RwoH · 2024-11-04

**Soundness:** 2
**Presentation:** 2
**Contribution:** 2
**Rating:** 6
**Confidence:** 4

**Summary:**

The paper titled "POROVER: IMPROVING SAFETY AND REDUCING OVERREFUSAL IN LARGE LANGUAGE MODELS WITH OVERGENERATION AND PREFERENCE OPTIMIZATION" presents a comprehensive study on enhancing the safety and reducing overrefusal in large language models (LLMs). The authors examine the impact of overgenerating training data using advanced teacher models on the safety and usefulness balance of instruction-following language models. They introduce POROver, a strategy that employs preference optimization methods to reduce overrefusal by leveraging completions from superior teacher models. The study demonstrates significant improvements in the F1 score between safety and usefulness, and a substantial reduction in overrefusal rates.

**Strengths:**

The paper introduces a novel approach to reducing over refusal in LLMs through overgeneration and preference optimization, which is a creative solution to a common problem in the field. The paper is well-written and the results are clearly presented, making it easy to follow the authors' reasoning and findings. The work addresses a critical issue in the deployment of LLMs, improving their safety without compromising their usefulness, which has significant implications for real-world applications.

**Weaknesses:**

- The paper primarily focuses on a single model size and family (Phi-3), which limits the generalizability of the findings. While the authors acknowledge this limitation, the lack of experimentation with different model scales makes it difficult to understand how these methods would perform across the spectrum of model sizes. This is particularly important given that safety and overrefusal behaviors often vary significantly with model scale. Including experiments with both smaller (3-4B) and larger (70B+) models would provide stronger evidence for the method's broad applicability. The paper's evaluation methodology relies heavily on automatic metrics and a limited set of benchmarks. While the chosen benchmarks (e.g., OR-Bench, XSTest) are relevant, they may not capture the full spectrum of real-world scenarios where safety and overrefusal matter. Including evaluations on more diverse datasets, particularly those featuring different languages, cultures, and domain-specific contexts, would strengthen the paper's conclusions about the method's effectiveness.

- The computational analysis of the proposed methods is notably absent from the paper. The overgeneration approach with GPT-4 as a teacher model likely incurs significant computational costs, yet there's no discussion of the training efficiency or resource requirements. This omission makes it difficult for practitioners to assess the method's feasibility in production environments. A detailed analysis of computational overhead compared to standard fine-tuning approaches would be valuable.

- The paper lacks a thorough comparison with existing safety and overrefusal reduction methods. While baseline comparisons are provided, the authors don't fully contextualize their results within the broader landscape of recent work on LLM safety alignment. A more comprehensive comparison with methods like constitutional AI, RLAIF, and other preference optimization approaches would better demonstrate the advancement over state-of-the-art.

- The robustness of the proposed method requires more thorough investigation. The paper doesn't examine how the method performs under adversarial conditions or when faced with edge cases. Additionally, there's no analysis of the consistency of results across multiple training runs or different random seeds. This makes it difficult to assess the reliability and stability of the approach in practice. The ethical implications of reducing overrefusal deserve deeper examination. While the paper successfully demonstrates technical improvements in reducing overrefusal, it doesn't adequately address the broader implications of making models more compliant.

**Questions:**

1. Can you provide more insight into why advanced teacher models require more safety examples? Is this related to the complexity of their responses or other factors?

2. How do you expect the observed trends to scale with different model sizes? Would smaller or larger models show similar patterns?

3. Could you elaborate on how different rejection sampling criteria were selected? Were other criteria considered?

   How sensitive are the results to the specific thresholds used in rejection sampling?

4. Could the authors expand on the ethical implications of their work, particularly regarding the balance between user freedom and model safety?

5. How do the results of POROver compare to other existing methods for improving LLM safety and reducing overrefusal? Are there any specific scenarios where POROver outperforms or falls short of other approaches?

6. Have you explored automated methods for tuning the containment threshold τ?

   Were other preference optimization methods considered besides DPO?

   How does the slight safety compromise in OR-Bench Toxic relate to the containment threshold?

---

> ### Author Response · Authors · 2024-11-28
>
> We thank the reviewer for their review and comments.
>
> We will answer the points raised individually.
>
> **W1, Q2: Exploration of different model families and sizes.**
>
> We agree that experimenting with different model families and sizes is crucial to achieve robust and convincing results. In response to this feedback, we have expanded our experiments to cover multiple model families and sizes. Our revised manuscript includes results for Llama-3.1-8B, Llama-3.2-3B, and Phi-3-7B. We present the results for Llama-3.1-8B in the main text and share the results of Phi-3-7B and Llama-3.2-3B in Appendix D2 and D3, respectively.
>
> While we observed subtle variations in the exact Not-Unsafe and Not-Overrefusal Rates across models during instruction finetuning, the comparative trends between older and newer teachers remained consistent. In addition, PORover effectively reduced overrefusal while maintaining safety across all tested models. Therefore, our conclusions remain consistent across all models we tested.
>
> We were not able to conduct experiments on models larger than 8B parameters due to computational resource constraints. We leave the exploration of scaling behavior with larger models as future work. In our revised manuscript, we have included this point in the Limitations and Future Work section.
>
> **W1: The paper's evaluation methodology relies heavily on automatic metrics and a limited set of benchmarks.**
>
> We thank the reviewer for this observation. Our evaluation prioritized XS-Test for human evaluation due to the large number of models being compared. While OR-Bench and XS-Test are the most relevant benchmarks for assessing overrefusal and safety, we acknowledge they may not capture all aspects. We have expanded our safety evaluation to additional datasets beyond XSTest and Or-Bench, though resource constraints prevented us from exploring benchmarks with higher variability such as different languages, cultures, and domain specific contexts.
>
> **W2: The resource requirements for overgeneration with GPT-4o as well as the training efficiency are absent.**
>
> Thank you for pointing this out. We have relied on GPT-4o’s API in order to perform overgeneration. GPT-4o’s API is roughly five times more expensive than GPT-3.5 (our baseline model family) according to [1]. We have added this discussion to Appendix C of our revised manuscript.
>
> Regarding the training costs, convergence times remained similar across different instruction finetuning datasets for the same base models. We have included this point to Appendix B of our revised manuscript.
>
>
> [1] OpenAI. “Pricing.” OpenAI, 2024, openai.com/api/pricing/.
>
>
> **W3: The paper lacks a thorough comparison with existing safety and overrefusal reduction methods. While baseline comparisons are provided, the authors don't fully contextualize their results within the broader landscape of recent work on LLM safety alignment.**
>
> We acknowledge the importance of situating our method, POROver, within the context of existing approaches like Constitutional AI, RLAIF, and preference optimization methods.
>
> •	Constitutional AI: While Constitutional AI uses a predefined set of principles to critique and refine outputs, POROver focuses on explicit overrefusal reduction using pairwise preference data. Unlike Constitutional AI, which relies on rule-based critiques, POROver targets the trade-off between safety and usefulness by leveraging advanced teacher model completions.
>
> •	RLAIF: POROver differs from RLAIF by avoiding the computational overhead of reinforcement learning and reward modeling. Instead, it uses DPO with curated preference data, simplifying the alignment process while achieving similar benefits.
>
> •	Preference Optimization: POROver innovates on standard DPO by generating preference datasets specifically for reducing overrefusal. This involves leveraging advanced safety scoring methods (e.g., Llama Guard 2) to improve usefulness while maintaining safety.
> By explicitly targeting overrefusal reduction, POROver fills a gap in existing methods that often prioritize safety at the cost of usefulness. We believe this demonstrates its complementary and novel contribution to the field.
>
> **W4: The paper doesn't examine how the method performs under adversarial conditions or when faced with edge cases.**
>
> While our models show major safety improvements for typical usage, they are not completely safe. Our benchmark results demonstrate good generalization, but the models may still be vulnerable to adversarial attacks and jailbreaking. Our work focuses on creating a layer of safety while maintaining usefulness. Although users might find ways to bypass safety measures, we believe that establishing safe behavior as the default is essential for user-facing applications. We have added these points to the Limitations and Future Work Section in our revised manuscript.

---

> > ### Author Response · Authors · 2024-11-28
> >
> > **W4: There's no analysis of the consistency of results across multiple training runs or different random seeds.**
> >
> > We acknowledge that the consistency analysis with multiple training runs and random seeds is critical for assessing the stability of the results. While we could not attend them due to resource and computational constraints, we have significantly expanded our experiments to cover two model families (Llama 3 and Phi-3) across various sizes. Our conclusions remain consistent across all model families and sizes we tested, providing robustness.
> >
> > **W4,Q4: Ethical implications of our work.**
> >
> > We believe that achieving the maximum level of safety is crucial in all applications. At the same time, high safety should not come at the cost of excessive overrefusal, which unnecessarily restricts legitimate user interactions. Importantly, the inverse - sacrificing safety measures to increase user freedom - is not an acceptable solution, as it could lead to harmful outcomes. Our work is an effort to maintain robust safety guardrails while preserving user freedom for appropriate requests, without compromising either aspect. This is essential for developing AI systems that are both protective and practical - ensuring safety without defaulting to overly conservative responses that could diminish the models' utility and accessibility. We have added this discussion to the Ethical Statement section and clarified our terminology throughout the revised manuscript.
> >
> > **Q1: Can you provide more insight into why advanced teacher models require more safety examples? Is this related to the complexity of their responses or other factors?**
> >
> > Thank you for this question. We think that the need for more safety examples stems from GPT-4o's more complex response patterns. GPT-4o generates noticeably longer and more complex responses compared to GPT-3.5, as shown in pendix C.1. This difference in response complexity may lead to nuanced safety signals during training.  We have added these points to Section 4.1 in our revised manuscript.
> >
> > **Q3: Could you elaborate on how different rejection sampling criteria were selected? Were other criteria considered?**
> >
> > Our selection of rejection sampling criteria was primarily guided by their established presence in the existing literature. While we explored several potential criteria during our initial experimental design phase, we ultimately focused on the presented set due to their widespread adoption and empirical validation in the literature. We have included the relevant citations in our revised manuscript. In the future, exploring more methods and reward models may provide novel insights for safety and usefulness. We have added this point to the Limitations and Future Work section in our manuscript.
> >
> > **Q3: How sensitive are the results to the specific thresholds used in rejection sampling?**
> >
> > The results were not vastly different except the two edge values: When τ = 0 (meaning no toxic prompts in preference training set), the model's safety performance dropped significantly for small gains of usefulness. We hypothesize that this occurred because the primary training signal becomes unconditional compliance with all prompts without toxic examples in the training set. When τ = 0.5, the model's usefulness stayed too low while high safety was maintained throughout the training. We have added this discussion to Section 4.2 in our revised manuscript.
> >
> > **Q5: How do the results of POROver compare to other existing methods for improving LLM safety and reducing overrefusal? Are there any specific scenarios where POROver outperforms or falls short of other approaches?**
> >
> > POROver is specifically designed to target models that are already highly safe but exhibit high overrefusal rates. Its primary goal is to enhance usefulness without compromising the existing safety of the model. This positions POROver as a complementary approach to safety fine-tuning or alignment techniques, which often focus on improving safety at the potential expense of usefulness.
> > To the best of our knowledge, no other post-training method specifically addresses reducing overrefusal while maintaining high safety in highly safe models. Methods like RLHF or other preference optimization algorithms focus broadly on alignment but may not explicitly mitigate overrefusal. POROver is novel in applying pairwise preference optimization to this specific problem using overgenerated data from superior teacher models.
> > Regarding limitations, POROver requires high-quality teacher model completions, which can be resource intensive. Additionally, while POROver shows promising results in reducing overrefusal in safe models, its application to less safe models remains unexplored. It is unclear whether POROver would offer advantages in such scenarios and investigating this is an interesting direction for future work.

---

> > > ### Author Response · Authors · 2024-11-28
> > >
> > > **Q6: Have you explored automated methods for tuning the containment threshold τ? Were other preference optimization methods considered besides DPO?**
> > >
> > > We acknowledge that there are opportunities to further enhance POROver. Due to computational constraints, we have not explored automated tuning of the containment threshold or investigated alternative optimization methods. We believe these directions hold significant promise for future work. Specifically, incorporating automated hyperparameter tuning tools and exploring reference-free preference optimization methods could potentially lead to more efficient implementations than the current approach. We have included these directions in our Limitations and Future Work section.
> > >
> > > **Q6: How does the slight safety compromise in OR-Bench Toxic relate to the containment threshold?**
> > >
> > > Our grid search over different containment thresholds showed that each threshold value leads to a slightly different point in the safety-usefulness trade-off curve. Based on this empirical finding, we believe that our explored values of tau might be suboptimal. In fact, this is closely tied to the need to explore automated methods for tuning the containment threshold. We have added this point to the Limitations and Future Work Section of our revised manuscript.
> > >
> > > Thank you very much for your valuable time and thoughtful review! We welcome any additional questions and are happy to provide further clarification.

---

> > > > ### Comment · Reviewer_RwoH · 2024-11-29
> > > >
> > > > Thank you for your detailed responses and revisions to address the concerns raised. While I appreciate the effort to improve the manuscript, I will maintain my current scoring due to the following key concerns:
> > > >
> > > > 1. Although the inclusion of multiple model families and sizes improves robustness, the lack of experiments with larger models (e.g., 70B+) significantly limits the generalizability of your conclusions. Understanding scaling behavior is crucial in assessing the broader applicability of your method, and this remains unexplored.
> > > >
> > > > 2. Expanding evaluations to additional datasets is noted, but the paper still does not adequately address the need for benchmarks featuring diverse languages, cultures, and domains. This limitation makes it difficult to assess the true real-world utility and robustness of the proposed method.
> > > >
> > > > 3. While you provide an estimate of GPT-4's API cost and training convergence times, the lack of detailed computational analysis still hinders practitioners' ability to evaluate the feasibility of implementing this approach in production.
> > > >
> > > > 4. The absence of experiments on adversarial scenarios or consistency across training runs remains a significant gap. These analyses are critical for evaluating the reliability and stability of the proposed method in practical applications.
> > > >
> > > > These points highlight critical areas where the paper still falls short, and I encourage further exploration and clarification in these aspects to strengthen the work. Thank you for addressing the other points comprehensively.

---

> > > > > ### Author Response · Authors · 2024-12-03
> > > > >
> > > > > We thank the reviewer for the valuable comments and feedback. We are happy to know that most of your concerns are addressed. We address your remaining concerns as follows:
> > > > >
> > > > > **Concern 1.** We expect the benefits of our methods to diminish as model size approaches that of the teacher models, since these larger models typically exhibit fewer safety and overrefusal issues. Our choice of model sizes aligns with standard practice in alignment research, where evaluation commonly focuses on 3B-8B models. While we acknowledge that testing on 70B+ models would provide additional insights, the computational costs and resource requirements make this impractical for our current study.
> > > > >
> > > > > **Concerns 2. and 4.** While our models demonstrate significantly improved safety during typical usage patterns, we acknowledge important limitations. Our experimental results show effective generalization across diverse instruction sets, and we contribute novel insights through evaluation on previously understudied overrefusal benchmarks. However, the models may still be vulnerable to edge cases involving unusual prompts, adversarial attacks, and jailbreaking attempts.
> > > > > Our work establishes a foundational safety layer that balances safety with usefulness. We recognize that determined users might bypass these safeguards, yet maintaining strong safety defaults remains crucial for many user-facing applications. Our approach creates meaningful friction against misuse while preserving the model's intended functionality.
> > > > >
> > > > > **Concern 3.** Given that we are using well-established training algorithms (SFT and DPO), we do not analyze their computational costs in our work. We believe that providing the relevant API costs and dataset sizes is sufficient for users to assess the feasibility of our approach for their use cases. We also note that the converge times do not vary between different training datasets, which would simplify their analysis.
> > > > >
> > > > > Thank you very much for your valuable time and thoughtful review! We welcome any additional questions and are happy to provide further clarification.

---

### Official Review · Reviewer_DgZU · 2024-11-04

**Soundness:** 3
**Presentation:** 4
**Contribution:** 3
**Rating:** 6
**Confidence:** 3

**Summary:**

This paper studies the impact of rejection sampling for safety training on the model's tendency to over-reject. The results show in the student teacher setting distilling from a stronger model like GPT-4 to Phi-3 the over-refusal reduces from near 100% to 45% on OR-bench.

**Strengths:**

1) The approach shows strong empirical improvement and scopes a relevant problem of over-refusing.

2) Although the results are shown only on the 7B Phi-3 model, it's done on a variety of seemingly-toxic datasets.

3) The results are supported by human annotations in appendix C.

4) The paper speaks to the trade off on the amount of safety training data needed to achieve the level of desired safety. This is defined in terms of additional safety datapoints.

**Weaknesses:**

1) The approach involves distilling from already safety trained models. In particular, safety trained models that are also likely targeting similar datasets. The work shows gpt3.5 vs gpt4, but it would be more convincing to show Llama-3 as the teachers also, or an somewhat unsafe teacher model.

2) Added Safety Data (ASD) is only evaluated at three levels 0, 2K, and 20K. More data would be needed to defend the claim that there is a tradeoff between ASD and safety. I would expect it to saturate based on the amount of base diversity represented in the prompts.

3) The figures have misleading axis starting at 85% to 100% for instance in Figure 4. This makes the difference look bigger than it is.

**Questions:**

1) Do these results replicate on a different model other than Phi-3? It would be especially convincing if it replicates on different sizes of the llama-3 family of models.

2) Is it possible that Phi-3 already has safety training that's particularly prone to over-refusing?

3) I interpret ASD to be the number of datapoints added after rejection sampling. Is this correct? If so, this correlates with the extent the model is deviated from the base model.

4) It's not clear where 15%, 45.5%, and 95.5% come from in the abstract.

---

> ### Author Response · Authors · 2024-11-28
>
> We thank the reviewer for their review and comments.
>
> We will answer the points raised individually.
>
> **W1, Q1: Exploration of different model families.**
>
> We agree that experimenting with different model families and sizes is crucial to achieve robust and convincing results. In response to this feedback, we have expanded our experiments to cover multiple model families and sizes. Our revised manuscript includes results for Llama-3.1-8B, Llama-3.2-3B, and Phi-3-7B. We present the results for Llama-3.1-8B in the main text and share the results of Llama-3.2-3B and Phi-3-7B in the Appendix D.3 and D.2, respectively. Given the established limitations of LLaMA models in terms of safety and usefulness [1][2][3], we strategically focused on using them as student models rather than teachers.
>
> About the robustness of our results, while we observed subtle variations in the exact Not-Unsafe and Not-Overrefusal Rates across different student models during instruction finetuning, the comparative trends between using older and newer teachers remained consistent. In addition, PORover effectively reduced overrefusal while maintaining safety across all tested models. Therefore, our conclusions remain consistent across all models we tested.
>
> [1] Bianchi, Federico, et al. “Safety-Tuned LLaMAs: Lessons from Improving the Safety of Large Language Models That Follow Instructions.” OpenReview, 2024, openreview.net/forum?id=gT5hALch9z.
>
> [2] Cui, Justin, et al. “OR-Bench: An Over-Refusal Benchmark for Large Language Models.” ArXiv.org, 2024, arxiv.org/abs/2405.20947.
>
> [3] Röttger, Paul, et al. “XSTest: A Test Suite for Identifying Exaggerated Safety Behaviours in Large Language Models.” ArXiv.org, 2023, arxiv.org/abs/2308.01263.
>
>
> **W2: Added Safety Data (ASD) is only evaluated at three levels 0, 2K, and 20K. More data would be needed to defend the claim that there is a tradeoff between ASD and safety. I would expect it to saturate based on the amount of base diversity represented in the prompts.**
>
> We agree that a fine-grained experiment would make the claim more convincing. We have expanded our initial ASD grid of {0, 2k, 20k} to {0, 2k, **5k**, **10k**, **15k**, 20k} for Llama-3.1-8B and evaluated its safety in our revised manuscript. As shown in Figure 5, the safety of the models increases with more ASD, and we see a saturation especially after 10k ASD, as you expected.
> Additionally, [1] has previously conducted fine-grained ASD analysis and showed the tradeoff between ASD and safety. We have added a citation to [1] to the related discussion in Section 4.3 our revised manuscript.
>
> [1] Bianchi, Federico, et al. “Safety-Tuned LLaMAs: Lessons from Improving the Safety of Large Language Models That Follow Instructions.” OpenReview, 2024, openreview.net/forum?id=gT5hALch9z.
>
> **W3: The figures have misleading axis starting at 85% to 100% for instance in Figure 4. This makes the difference look bigger than it is.**
>
> Thank you for pointing this out. We have replaced the figure causing confusion with a table (Table 2) in our revised manuscript. We have also revised the discussion in Section 4.1 to clearly communicate that the results are similar in Table 2.
>
> **Q2: Is it possible that Phi-3 already has safety training that's particularly prone to over-refusing?**
>
> In our revised manuscript, we investigate both Phi-3 and the Llama family of models. Our expanded analysis shows similar overrefusal trends across these model families, suggesting this behavior is not unique to Phi-3.
>
> **Q3: I interpret ASD to be the number of datapoints added after rejection sampling. Is this correct? If so, this correlates with the extent the model is deviated from the base model.**
>
> Yes, we define ASD as the number of toxic prompts added to the training set. The completions for these toxic prompts are overgenerated with GPT-4o and then rejection sampled. We agree that rejection sampling is a determining factor on how much the model is deviated from the base model.
>
> **Q4: It's not clear where 15%, 45.5%, and 95.5% come from in the abstract.**
>
> Thank you for pointing this out. We have changed those overrefusal rates to Not-Overrefusal Rates (of Llama-3.1-8B) in our revised manuscript to increase the connection between the Abstract and the Results section.
>
> Thank you very much for your time and review! We would greatly appreciate any additional questions and are happy to provide further clarification.

---

> > ### Author Response · Authors · 2024-12-01
> >
> > Dear reviewer DgZU,
> >
> > Thank you again for your time! Since the discussion period ends tomorrow, we just wanted to see if our response has clarified your questions. We hope you would consider increasing your score if we have answered your questions. Please let us know if you have additional comments and we are happy to follow up. Thanks!

---

### Official Review · Reviewer_PDLs · 2024-11-08

**Soundness:** 2
**Presentation:** 2
**Contribution:** 2
**Rating:** 5
**Confidence:** 4

**Summary:**

This paper is concerned with training language models that output safe content but do not refuse too often. They test two algorithmic techniques to achieve this. First, they use overgeneration, which involves sampling multiple possible outputs and choosing the best responses for training. Second, they generate preference data pairs, based on responses that were unsafe/over-refusal vs not.

**Strengths:**

1. The paper tackles an important problem of making models safer.
2. The paper evaluates on multiple benchmarks with different step counts to give broader analysis.
3. The paper's algorithm seems straightforward to implement.

**Weaknesses:**

1. I believe the algorithms in this work have limited novelty. Rejection sampling and preference optimization are some of the most used tools for current fine-tuning and safety alignment, so the paper needs to provide novel analysis instead.
2. I'm confused about the empirical gains. It seems that in Table 1, the random selection GPT-4o baseline performs on par with the rejection sampling, indicating that the filtering step is not that crucial. Moreover, in Figures 3 and 4, training on GPT-3 seems to be extremely safe (though it does not solve over-refual).

In general, I suggest focusing on key empirical takeaways, ensuring that POROver improves upon simple baselines, and organizing the presentation of the results.

**Questions:**

1. Looking at Figure 5, it looks like there was little improvement on XSTest performance and a reduction in over-refusal for ORBench but no improvement in safety. Why do you think this is? Is it related to the training set being ORBench?

---

> ### Author Response · Authors · 2024-11-28
>
> We thank the reviewer for their review and comments.
>
> We will answer the points raised individually.
>
> **W1: I believe the algorithms in this work have limited novelty. Rejection sampling and preference optimization are some of the most used tools for current fine-tuning and safety alignment, so the paper needs to provide novel analysis instead.**
>
> Thank you for pointing this out. While we acknowledge that rejection sampling and preference optimization are commonly used in other domains, our goal is investigating their use to obtain novel insights about safety and usefulness. To the best of our knowledge, no other work has presented a quantitative and systematic analysis of safety and usefulness when comparing older and newer, more superior teacher models during instruction fine-tuning with general-purpose and toxic prompts. Additionally, we are not aware of any work in the literature that specifically targets over-refusal reduction while maintaining safety. We have revised the Introduction section in our manuscript and added the following motivations and clarifications about our novelties:
>
> 1.	While using advanced teacher models (e.g., GPT-4o) for instruction fine-tuning with general-purpose prompts is known to enhance a student's general capabilities, we systematically analyze its previously unexplored impact on safety and usefulness. We find that it significantly enhances the model's safety and usefulness balance. Our models demonstrate significantly increased safety with only a modest reduction in usefulness.
> 2.	The few available open-source instruction fine-tuning datasets containing toxic prompts present a significant challenge: they lead to high overrefusal in trained models. Models trained on these datasets tend to develop significant overrefusal in their attempt to achieve the highest safety levels [1]. Notably, these datasets were generated using older models like GPT-3.5 as teachers. In our work, we investigate the impact of using more recent, advanced models to generate training data for safety finetuning. Our results reveal that models trained with completions generated by superior teacher models develop significantly less overrefusal. However, obtaining high safety levels with superior teacher models requires larger training datasets, revealing a previously undocumented trade-off in safety assurance and data efficiency.
> 3.	There are numerous recent LLMs that are highly safe but exhibit high overrefusal, including Claude-3, Gemini-1.5, Llama-2, and Llama-3 [2][3]. While this behavior may stem from conservative safety filtering during training, the exact mechanisms remain unclear due to the proprietary nature of training datasets and procedures. In scenarios where a model is highly safe but exhibits high overrefusal, the goal becomes reducing over-refusal while maintaining the high safety level. We introduce POROver to specifically target this scenario. To the best of our knowledge, our paper is the first to address the goal of reducing overrefusal while maintaining high safety, and to evaluate the use of preference optimization methods specifically targeted to this goal.
>
> We believe that these contributions, all supported by quantitative evidence through standard open-source benchmarks, provide a comprehensive framework for understanding and optimizing safety and usefulness in language models, establishing novel insights. Given the critical importance of safety, we have also taken steps to address the lack of high-quality open-source finetuning datasets by releasing all the data we generated with GPT-4o.
>
>
> [1] Bianchi, Federico, et al. “Safety-Tuned LLaMAs: Lessons from Improving the Safety of Large Language Models That Follow Instructions.” OpenReview, 2024, openreview.net/forum?id=gT5hALch9z.
>
> [2] Cui, Justin, et al. “OR-Bench: An Over-Refusal Benchmark for Large Language Models.” ArXiv.org, 2024, arxiv.org/abs/2405.20947.
>
> [3] Röttger, Paul, et al. “XSTest: A Test Suite for Identifying Exaggerated Safety Behaviours in Large Language Models.” ArXiv.org, 2023, arxiv.org/abs/2308.01263.

---

> > ### Author Response · Authors · 2024-11-28
> >
> > **W2: I'm confused about the empirical gains. It seems that in Table 1, the random selection GPT-4o baseline performs on par with the rejection sampling, indicating that the filtering step is not that crucial.**
> >
> > Thank you for this observation. We would like to first note that we have expanded our experiments to cover multiple model families and sizes to enhance the robustness and generalizability of our findings. Our revised manuscript includes results for Llama-3.1-8B, Llama-3.2-3B, and Phi-3-7B. We present the results for Llama-3.1-8B in the main text and share the results of Phi-3-7B and Llama-3.2-3B in Appendix D2 and D3, respectively since our findings remain consistent. Therefore, Table 1 shows the results for Llama-3.1-8B in our revied manuscript, and Phi-3-7B results are in Table 6.
> >
> > While random selection and rejection sampling may appear similar at first glance, our results reveal that rejection sampling effectively identifies safer operating points while preserving model usefulness, avoiding unnecessary trade-offs between safety and usefulness. For instance, in OR-Bench, when using the ArmoRM helpfulness criterion:
> >
> > 1.	Phi-3-7B's F1-score on improves by 2.75%, driven by enhancements in both Not-Unsafe Rate and Not-Overrefusal Rate (Table 6)
> >
> > 2.	Llama-3.1-8B's F1-score increases by 1.02% while its safety increases by 5.65% (Table 1)
> >
> > 3.	Llama-3.2-3B shows a 0.51% improvement in F1-score while its safety increases by 1.07% (Table 7)
> >
> > These consistent improvements across different models support our findings. We have elaborated on these points in Section 4.1 in our revised manuscript and added a detailed discussion in Appendix E.
> >
> > **W2: Moreover, in Figures 3 and 4, training on GPT-3.5 seems to be extremely safe (though it does not solve over-refusal).**
> >
> > Thank you for this observation. The extremely safe but overrefusing behavior of GPT-3.5 highlights the key weakness of older teacher models. Training on GPT-3.5 gives high safety with the cost of a high overrefusal. However, training on GPT-4o achieves the same high safety level with much less overrefusal. Therefore, we can conclude that using better teacher models effectively reduces the development of overrefusal during safety finetuning.
> >
> > We have revised the discussion of these findings in Section 4.1 to communicate these points more clearly.
> >
> > **W3: Ensuring that POROver improves upon simple baselines.**
> >
> > We appreciate the concern about baseline comparisons. As we have elaborated in our reply to Weakness 1, we are not aware of any prior work specifically targeting over-refusal reduction in already-safe LLMs – we present this as novel problem space distinct from general safety improvements. Our baselines are before-POROver versions of each tested model, which allows us to directly measure the impact of our method.
> > Our experiments now include comprehensive evaluations across multiple models (Llama-3.1-8B, Llama-3.2-3B, and Phi-3-7B), demonstrating consistent improvements. Speicifically, in OR-Bench:
> >
> > 1.	Llama-3.1-8B improves its usefulness from 57.6% to 82.1%.
> > 2.	Phi-3-7B improves its usefulness from 54.8% to 85%.
> > 3.	Llama-3.2-1B improves its usefulness from 54.8% to 83.4%.
> >
> > All models achieve these improvements with less than 1% drop in their safety. We believe these results effectively demonstrate POROver's value compared to the baseline performance of unmodified models.
> >
> > **W3: Focusing on key empirical takeaways and and organizing the presentation of the results.**
> >
> > Thank you for raising this point. We have reorganized our Results section to highlight the key findings more clearly. We have also improved our discussions throughout the manuscript to better explain our main insights and empirical results.

---

> > > ### Author Response · Authors · 2024-11-28
> > >
> > > **Q1: Looking at Figure 5, it looks like there was little improvement in XSTest performance and a reduction in over-refusal for ORBench but no improvement in safety. Why do you think this is? Is it related to the training set being ORBench?**
> > >
> > > We would like to note that we show an ablation analysis in Figure 5 in our revised manuscript. The before- and after-POROver results for Llama-3.1-8B are in Figure 4 and the results for Phi-3 are presented in Figure 10. Both figures demonstrate that POROver achieves its primary goal: reducing over-refusal while maintaining the model's existing safety levels.
> > >
> > > The smaller gains in XSTest's Not-Overrefusal Rate compared to OR-Bench can be explained by ceiling effects - the base model was already performing well on XSTest (92.8% Not-Overrefusal Rate for Phi-3-7B), leaving limited room for improvement. We suspect that this is because XSTest is a smaller, older benchmark with less diversity. In contrast, Phi-3-7B started at just 54.8% Not-Overrefusal Rate on OR-Bench, giving us more headroom to demonstrate improvement - which we achieved by reaching 85%. We note that we observe consistent improvements across all tested Llama models, confirming that these patterns are not specific to any single model architecture.
> > >
> > > We have revised Section 4.2 to communicate these points more clearly.
> > >
> > > Thank you very much for your valuable time and thoughtful review! We welcome any additional questions and are happy to provide further clarification.

---

> > > > ### Author Response · Authors · 2024-12-01
> > > >
> > > > Dear reviewer PDLs,
> > > >
> > > > Thank you again for your time! Since the discussion period ends tomorrow, we just wanted to see if our response has clarified your questions. We hope you would consider increasing your score if we have answered your questions. Please let us know if you have additional comments and we are happy to follow up. Thanks!

---

> > > > > ### Comment · Reviewer_PDLs · 2024-12-03
> > > > >
> > > > > I find the clarification of the work's intention to be "reducing over-refual while preserving original safety capabilities" helpful, since it seems like there aren't strong improvements. I also understand the novelty in the problem space of reducing refusals. I increase my score to 5 since I still feel like the method is not conceptually different from other fine-tuning algorithms and that the gains/scope are somewhat limited.

---

> > > > > > ### Author Response · Authors · 2024-12-03
> > > > > >
> > > > > > We thank the reviewer for the valuable comments and feedback. We are happy to know that our revised manuscript is helpful. We address your remaining concerns as follows:
> > > > > >
> > > > > > **1. Method is not conceptually different from other fine-tuning algorithms:**
> > > > > >
> > > > > > We acknowledge the importance of situating POROver within the context of existing approaches like Constitutional AI, RLAIF, and preference optimization methods.
> > > > > >
> > > > > > • Constitutional AI: While Constitutional AI uses a predefined set of principles to critique and refine outputs, POROver focuses on explicit overrefusal reduction using pairwise preference data. Unlike Constitutional AI, which relies on rule-based critiques, POROver targets the trade-off between safety and usefulness by leveraging advanced teacher model completions.
> > > > > >
> > > > > > • RLAIF: POROver differs from RLAIF by avoiding the computational overhead of reinforcement learning and reward modeling. Instead, it uses DPO with curated preference data, simplifying the alignment process while achieving similar benefits.
> > > > > >
> > > > > > • Preference Optimization: POROver innovates on standard DPO by generating preference datasets specifically for reducing overrefusal. This involves leveraging advanced safety scoring methods (e.g., Llama Guard 2) to improve usefulness while maintaining safety. By explicitly targeting overrefusal reduction, POROver fills a gap in existing methods that often prioritize safety at the cost of usefulness. We believe this demonstrates its complementary and novel contribution to the field.
> > > > > >
> > > > > > **2. The gains/scope are somewhat limited.**
> > > > > >
> > > > > > POROver addresses an under-explored but significant challenge: reducing overrefusal in models that are already highly safe. This issue affects many prominent models including Claude-3, Gemini-1.5, Llama-2, and Llama-3. While existing safety fine-tuning and alignment techniques often improve safety at the cost of increased overrefusal, POROver takes a complementary approach by optimizing model usefulness while maintaining safety levels. Given the widespread occurrence of overrefusal across major language models, our method has broad practical applications.
> > > > > >
> > > > > > Thank you very much for your valuable time and thoughtful response! We welcome any additional questions and are happy to provide further clarification.

---

### Official Review · Reviewer_h5oh · 2024-11-12

**Soundness:** 3
**Presentation:** 3
**Contribution:** 2
**Rating:** 6
**Confidence:** 3

**Summary:**

The authors present a framework that aims to reduce overrefusal in Large Language Models (LLMs), while improving their safety. It involves finetuning on overgenerated training data from teacher models, such as GPT-4o, and preference optimization to guide models to respond in benign (but possibly seemingly toxic) prompts. Through experiments on Phi-3 7B, and various teacher models, the authors find that their method achieves significant reduction in overrefusal, while maintaining a balance between usefulness and safety.

**Strengths:**

* The paper tackles an important aspect of LLMs, aiming to investigate and improve their tradeoff in mainting (or even improving) their safety, without, however, undermining their usefulness due to overrefusal.
* The experiments presented are extensive; the effectiveness of the presented method has been evaluated on a variety of datasets and benchmarks related to safety and overrefusal.
* The experiments suggest that the proposed framework effectively results in a balance between usefulness and safety, without significantly undermining the general capabilities of the tested model.

**Weaknesses:**

* As the authors acknowledge, a limitation of their study is that the proposed framework is only tested on a single model family and size (e.g., Phi-3 7B). In my opinion, while the results are promising, this limitation is significant; given that the framework relies on finetuning and preference optimization of pretrained models, testing it across diverse model families and scales would prove its effectiveness and generality. It is unclear to me whether the results would be similar in that case.
* Adding more fine grained experiments on the number of Added Safety Data (ASD) would make the claim that **the proposed method is effective without undermining the general abilities of the tested model** more convincing.

**Questions:**

* Although experiments on models with different scales were not included, how do you expect the models to behave, assuming that they come from the same family? Would the benefits saturate as the number of parameters increases?
* How sensitive is the proposed method in the choice of the hyperparameter $\tau$? Were the results vastly different accross your grid search?

---

> ### Author Response · Authors · 2024-11-28
>
> We thank the reviewer for their review and comments.
>
> We will answer the points raised individually.
>
> **W1, Q1: Exploring multiple model families and sizes.**
>
> We agree that experimenting with different model families and sizes is crucial to achieve robust and convincing results. In response to this feedback, we have expanded our experiments to cover multiple model families and sizes. Our revised manuscript includes results for Llama-3.1-8B, Llama-3.2-3B, and Phi-3-7B. We present the results for Llama-3.1-8B in the main text and share the results of Phi-3-7B and Llama-3.2-3B in Appendix D2 and D3, respectively.
>
> While we observed subtle variations in the exact Not-Unsafe and Not-Overrefusal Rates across models during instruction finetuning, the comparative trends between older and newer teachers remained consistent. In addition, PORover effectively reduced overrefusal while maintaining safety across all tested models. Therefore, our conclusions remain consistent across all models we tested.
>
> We were not able to conduct experiments on models larger than 8B parameters due to computational resource constraints. We leave the exploration of scaling behavior with larger models as future work. In our revised manuscript, we have included this point in the Limitations and Future Work section.
>
> **W2: Adding more fine-grained experiments on the number of Added Safety Data (ASD) would make the claim that the proposed method is effective without undermining the general abilities of the tested model more convincing.**
>
> We agree that a fine-grained experiment would make the claim more convincing. We have expanded our initial ASD grid of {0, 2000, 20000} to {0, 2000, **5000**, **10000**, **15000**, 20000} for Llama-3.1-8B and evaluated its general capabilities in our revised manuscript. As shown in Figure 6, the general capabilities of the models remained consistent.
> Additionally, [1] has previously conducted a similar fine-grained ASD analysis and discussed that the general capabilities don’t get affected by ASD up to a certain ASD level. Our results show that we were within that acceptable range in our experiments. We have added a citation to [1] to the related discussion in Section 4.3 of our revised manuscript.
>
> [1] Bianchi, Federico, et al. “Safety-Tuned LLaMAs: Lessons from Improving the Safety of Large Language Models That Follow Instructions.” OpenReview, 2024, openreview.net/forum?id=gT5hALch9z.
>
> **Q2: How sensitive is the proposed method in the choice of the hyperparameter? Were the results vastly different across your grid search?**
>
> The results were not vastly different except the two edge values: When τ = 0 (meaning no toxic prompts in preference training set), the model's safety performance dropped significantly for small gains of usefulness. We hypothesize that this occurred because the primary training signal becomes unconditional compliance with all prompts without toxic examples in the training set. When τ = 0.5, the model's usefulness stayed too low while high safety was maintained throughout the training. We have added this discussion to Section 4.2 in our revised manuscript.
>
> Thank you very much for your time and review! We would greatly appreciate any additional questions and are happy to provide further clarification.

---

> > ### Comment · Reviewer_h5oh · 2024-12-01
> >
> > Thank you for your response.
> > Your additional experiments and discussion address most of my major concerns.
> > Thus, I have increased my overall score, and my soundness and presentation scores.
> >
> > Below are some minor additional comments:
> >
> > * After moving the Phi-3 results to the appendices, I noticed a few minor figure reference errors in your draft. For example, around line ~466, it seems you should reference Figure 4 instead of Figure 11.
> >
> > * I suggest including the model’s name (Llama-3.1 8B) in all figures in the main text, as you have done in the appendices, to improve clarity and make the paper easier to follow.

---

> ### Author Response · Authors · 2024-12-03
>
> We thank the reviewer for the valuable comments and feedback. We are happy to know that most of your concerns are addressed. We will consider their formatting suggestions in our final manuscript. As a note about the larger models, we expect the benefits of our methods to diminish as model size approaches that of the teacher models, since these larger models typically exhibit fewer safety and overrefusal issues. Our choice of model sizes aligns with standard practice in alignment research, where evaluation commonly focuses on 3B-8B models. While we acknowledge that testing on 70B+ models would provide additional insights, the computational costs and resource requirements make this impractical for our current study.
>
> Thank you very much for your valuable time and thoughtful review! We welcome any additional questions and are happy to provide further clarification.

---

### Meta-Review · Area_Chair_iTsD · 2024-12-16

**Metareview:**

This paper aims to understand the beenefits of over generation, and forms of finetuning/rejection sampling to improve usefulness/safety tradeoffs in LLMs. While authors agreed that this is an important direction of study, many felt that it was somewhat incremental compared to prior work. Initial experiments focused only on the Phi models, and while later experiments include Llama models, I am still inclined to hold the limited scope against the authors.

Some reviewers brought up concerns that the authors looked at models with too few parameters. While indeed this may affect validity of the finding, enforcing stringent guidelines on model size for submissions unfairly disadvantages researchers with smaller compute budgets. Nevertheless, given that the contribution of the paper is on the evaluation side more than novelty, I believe that the authors do need to be more extensive and detailed in their experimentation. From my own reading, I also found it difficult to discern their precise methodology from reading the submission, and there was no pseudocode to be found. Ultimately, with more novelty, more breadth, and improved presentation, this work could be a nice contribution. But where it stands, I believe the work has room to improve before meriting acceptance.

**Additional Comments On Reviewer Discussion:**

While reviewer discussion was less thorough than I would have appreciated, the most vocal reviewer advocated for rejection. Their concerns included small model sizes and lack of culture/language diversity (I think this is acceptable if we wish to allow for compute-restricted academic research budgets), but the concerns about poor method documentation and lack of robustness to adversarial attacks were compelling.

---

### Decision · Program_Chairs · 2025-01-22

Reject